# Creating a Material Spectral Library for Plaster and Mortar Material Determination

**DOI:** 10.3390/ma14227030

**Published:** 2021-11-19

**Authors:** Eva Matoušková, Karel Pavelka, Saleem Ibrahim

**Affiliations:** Department of Geomatics, Faculty of Civil Engineering, Czech Technical University in Prague, Thákurova 7, Praha 6, 166 29 Prague, Czech Republic; pavelka@fsv.cvut.cz (K.P.); ibrahim@fsv.cvut.cz (S.I.)

**Keywords:** reflectance spectroscopy, spectral library, material decomposition

## Abstract

Historic object analysis and the knowledge of composition play an important role in restoration processes. Based on this information, restoration works are conducted. This paper introduces a non-invasive technique of plaster and mortar material decomposition using reflectance spectroscopy. For this purpose, a NIRQuest512-2,5 from Ocean Optics^®^/Ocean Insight^®^, is used to create a unique spectral library consisting of various materials. They were carefully selected to include those that were and still are commonly used for a plaster and mortar production. Each material of the spectral library was mapped in detail, verified using scanning electronic microscope (SEM) data, and the results were compared to a previously determined spectral signature. The new spectral library was then tested on 11 unknown plaster and mortar samples and verified using a scanning electronic microscope. It was found that reflectance spectroscopy provides a powerful tool for plaster and mortar material decomposition, although at the moment it cannot fully replace invasive techniques like chemical analyses or other invasive techniques. It provides relevant information that can be used for restoration works.

## 1. Introduction

Determination of plaster and facade composition is a very important issue in the restoration process. The composition can help with date assessment of the building and it can also provide information about restoration processes during past years. It is desired to use non-invasive methods for these analyses since no harm will be done to the plaster or facade [1,2]. Non-invasive methods have to be verified in the laboratory before their use and a detailed methodology has to be followed.

Literature shows various methods for plaster and facade documentation and analysis. Two non-invasive methods are commonly used, X-ray diffraction and Raman spectroscopy. When using X-ray diffraction, a monochromatic X ray-beam goes through the substance and beam diffraction occurs. The direction and intensity of these diffracted beams depend on an inner sample structure. It allows us to define the absolute molecule structure of the sample. This method can be used for identification of remains [3], for concrete erosion determination [4] or when combined with other methods for plaster analysis [5,6]. Raman spectroscopy is based on the Raman phenomenon (interaction between photons and light). The Raman scattering is based on the inelastic scattering of photons by matter, meaning that there are both an exchange of energy and a change in the light’s direction. A laser beam is shot onto a surface, it interacts with electrons, and then a photon is emitted. The wavelength of this photon is measured and it provides information about the sample. This method is used in particular for pigment determination of historical objects in Romania [7] and Mexico [8] or for efflorescence (soil migration) mapping in concrete [9].

Different surface features reflect and absorb the electromagnetic radiation of the sun in different ways. The reflectance properties of an object depend on the material and its physical and chemical state, the surface roughness as well as the angle of the sunlight. The reflectance of material also varies with the wavelength of the electromagnetic radiation. The amount of reflectance from a surface can be measured as a function of wavelength range, this is referred to as spectral reflectance. Spectral reflectance is a measure of how much energy (as a percent) a surface reflects at a specific wavelength. Surfaces reflect a different amount of energy in different portions of the spectrum. These differences in reflectance make it possible to identify different surface features or materials by analyzing their spectral reflectance signatures. Spectral reflectance curves graph the reflectance (in percent) of objects as a function of wavelengths [10]. Some materials reflect radiation of a certain wavelength, while other materials absorb the radiation of the same wavelength. These patterns of reflectance and absorption depending on wavelengths can uniquely identify certain materials. Reflectance spectroscopy has been already used for differentiating between anthropogenic calcite in plaster, ash, and natural calcite [11]. In this paper, the distinction is made using absorption peaks. In combination with other spectroscopic methods, it was used for pigment identification [12]. Three freely available spectral libraries provide information about building materials—London Urban Micromet data Archive—the Spectral Library of impervious Urban Materials (LUMA-SLUM) [13], Karlsruhe Library of Urban Materials [14] and Version 1.0 of the ECOSTRESS spectral library which was released on 2 February 2018 [15]. These libraries are very powerful, but none of them is allocated to Central Europe or the Czech Republic, respectively. These libraries were tested and similar samples were compared. However, it was found that they are not suitable for the location of interest since common local materials are always used for mortar and plaster manufacturing.

The facade is an external wall of the building and its final treatment that protects the building from weather conditions. It is very important for a building’s lifecycle; if it is harmed this can cause significant damage to the building. To secure building walls, plasters are commonly used. Plaster is a covering layer that is applied to a building inside and outside walls [16]. It creates an even, smooth surface and conceals the bumpiness of masonry. During history, plaster materials developed rapidly and the diversification is distinguishable worldwide [17,18]. Due to its everyday use, it is usually composed of materials that are ordinarily in a given location and their composition is analyzed by various research teams, e.g., [19]. Due to its composition nature, one can divide these materials into two types—lime-based (Europe) and gypsum-based (Middle East).

## 2. Materials and Methods

### 2.1. Reflectance Spectroscopy Measuring Device

A spectrometric measuring device consists of a spectrometer, illumination source, laptop with controlling software, fiber optics and a measuring probe, (Figure 1). The spectrometer NIRQuest512-2.5 from Ocean Optics^®^/Ocean Insight^®^, Orlando, FL, USA [20] works in the 900–2500 nm spectral range. In this range, 512 spectral bands are detectable. This spectrometer was chosen due to its spectral range following the previously purchased hyperspectral scanning device VNIR Hyperspec A-series, Bolton, MA, USA working between 400 and 1000 nm. This spectrometer can easily be used for laboratory and in situ measurements and its purchase cost was acceptable, although its main limitation is the restricted spectral range. More on this subject can be found in the Discussion chapter. The spectrometer is equipped with an InGaAs (Indium Gallium Arsenide) detector and cooling that is a necessity when using these detectors. Illumination of the object of interest is undertaken by using fiber optics. An external illumination source Cool Red provides an adequate light source for the 900–2500 nm spectral range. A multimodal optic fiber (permeability 400–2100 nm) with 600 um core diameter then transfers the light into the scanning probe (Figure 2). Relative intensity of the Cool Red illumination source and its dependence on the wavelength graph has been added (Figure 3). It is used for relative measurements hence the intensity is defined in percentage. A spectroscopic measuring device consists also of a laptop with control software, fiber optics and a measuring probe. The probe [21] is composed of an optical fiber that transfers information into the spectrometer as well as fibers bringing illumination from an external source. The end of the probe can be seen in Figure 2 and the ferrule diameter is 3.175 mm and with an acceptance angle of 24.8°. White reference material is a key issue for spectroscopic measurements. After several tests, it was found that Spectralon^®^ from LabSphere, North Sutton, NH, USA [22], will be used as a white reference. Data from a reflectance spectrometer are usually given in the form of a text (ASCII) file or a table. There is information about a signal that is captured by a detector for every spectral band in an operation spectral range. These data are then processed using an image processing software tool, in this case using ENVI, in scripts written in programming languages like C++ or programming software packages as Matlab R2019a.

### 2.2. Data Verification—Scanning Electron Microscope Analysis

Scanning electron microscope analysis was used to verify and evaluate spectroscopy results. Every material sample was analyzed in a laboratory and element composition data were obtained. This was based on energy-dispersive microscopy (EDS) that is a standard procedure for identifying and quantifying the composition of sample areas of a micron or less. The characteristic X-rays are produced when a material is bombarded with electrons in an electron beam instrument [23].

Laboratory measurements were conducted by the Department of Mechanics, Faculty of Civil Engineering, Czech Technical University in Prague [24]. A Phenom XL Desktop microscope [25] from the Nanoscience instrument company, Phoenix, AZ, USA that is available at the Department of Mechanics, FCE, CTU was used in a similar way to previous projects [26,27].

SEM results were then transferred to the form of a graph to show the representation of individual elements. A text overview is also presented for every material and plaster/mortar sample.

## 3. Measurement and Data Processing

### 3.1. Material Spectral Library Creation

In order to analyze material samples from Central Europe, a CTU Material Spectral Library was created. The main goal of this library is to provide spectroscopic information about materials that were used to produce historical and up to date plasters and mortars. This library is available after a request for all interested individuals.

For the spectral library creation, a CTU reflectance spectroscopy measuring device was used. It is an OceanOptics/OceanInsight, Orlando, FL, USA NIRQuest spectrometer with a reflectance probe; for more information see Section 2.1.

The spectral library was created in 2020 and it consists of 20 materials that are commonly used for plaster and mortar production in the area of Central Europe. This set was created and delivered by the Institute of Theoretical and Applied Mechanics of the Czech Academy of Sciences, Department of Material Research that has long-term experience with plaster and mortar composition determination [28,29]. It provides material spectral information as well as their analysis using a scanning electron microscope. This library is freely available upon request, and please contact the corresponding author for more information or see http://lfgm.fsv.cvut.cz/CTU.html (accessed on 14 November 2021).

Due to the large number of materials only two examples are presented in this paper (Section 3.2 and Section 3.3). All analyzed materials can be found at CTU DSpace library (http://hdl.handle.net/10467/94180, accessed on 14 November 2021).

### 3.2. Example of Known Material Included in the Spectral Library—Hořice Sandstone

#### 3.2.1. Sample Information

Hořice sandstone is quarried on the southern hillside of Hořice ridge close to the town of Podhorní Újezd, Czech Republic, about 100 km north-east of Prague, by the Kámen Ostroměř company, Ostroměř, Czech Republic. The mining area was created by joining three original quarries and it is therefore relatively extensive. Hořice sandstone is small-grained with a light ochre base colored with a copious but very delicate ferric pattern.

#### 3.2.2. Spectroscopy Measurement

The sample was documented using an ordinary mobile phone camera (Figure 4) and a digital DinoLite microscope (AnMo Electronics Corporation, Taipei, Taiwan) with magnification 50× (Figure 5) and 200× (Figure 6).

Acquired reflectance spectroscopy data were processed and analyzed using a MATLAB script. Analysis results are mentioned below, Figure 7 and Figure 8 show spectral graphs of the sample.

Minimum standard deviation: 1.32% (for λ = 1816.356 nm)

Maximum standard deviation: 2.99% (for λ = 1001.238 nm)

Mean standard deviation: 1.69%

#### 3.2.3. Scanning Electronic Microscope Findings

The sample was deluged using epoxy resin (*C_21_H_25_ClO_5_)*, sanded using *SiC* papers #2000, #4000 with water, and dusted. The sample photos (Figure 9, Figure 10 and Figure 11) show mainly *SiO_2_* grains—light grey color, and big pores—black color. Particle size varies from 70 to 300 um. Dark grey color can be seen in between grains, pores are filled with epoxy resin (*C_21_H_25_ClO_5_)*. White grains contain *Al*, *K*, *Mg* apart from *Si*.

SEM results (average atomic concentration, average weight concentration and average stoichiometry weight concentration) in the form of graphs are shown in Figure 12, Figure 13 and Figure 14.

### 3.3. Determination of Unknown Sample—Sample D, Čachtice

The processing workflow is shown in Figure 15, and this shows how an unknown sample is analyzed. After its spectral signature is determined and scanning electronic microscope analysis conducted, this sample is compared to a previously created spectral library of known samples using five different algorithms—spectral angle mapper (SAM), spectral information divergence (SID), non-negative least squares method (NNLS), spectral feature fitting (SFF) and binary encoding (BE).

SAM is a physically-based spectral classification that uses an n-D angle to match pixels to reference spectra. The algorithm determines the spectral similarity between two spectra by calculating the angle between the spectra and treating them as vectors in a space with dimensionality equal to the number of bands [30]. This tool was used in two software packages, ENVI 5.5 and Matlab R2019a, where a “hypersam” script was used and can be downloaded from MATLAB Hyperspectral Toolbox [31].

SID is a spectral classification method that uses a divergence measure to match pixels to reference spectra. The smaller the divergence, the more likely the pixels are similar [32]. This tool was used in Matlab since it is not available in ENVI’s Spectral Analyst and is available for spatial data only. MATLAB “hypersid“ script was used.

SFF compares the unknown spectra and a reference spectrum using a least-squares technique. SFF is an absorption-feature-based methodology. The reference spectra are scaled to match the image spectra after the continuum is removed from both datasets [33]. This method has been used in ENVI software package in the Spectral Analyst toolbox [34].

The binary encoding technique encodes the data and endmember spectra into zeros and ones, based on whether a band falls below or above the spectrum mean, respectively. This method has been used in the ENVI software package in the Spectral Analyst toolbox [34]. For each algorithm a ranking is created, and the final result is an average of all used algorithms.

This algorithm is included in the QSdata 0.9.4. open-source software [35] created by Prof. Ing. Aleš Čepek, CSc. It is based on a non-negative least squares algorithm [36,37] and seeks non–negative linear coefficients whose sum is equal to one. This software package is also suitable for average spectral curve calculations, although a standard deviation of the result is missing. This issue was discussed and is planned to be added shortly. All information and abilities of this powerful open-source software tool are available online [35].

#### 3.3.1. Sample Information

Eleven unknown samples were researched under the names 1–4, A–F and one rock sample. Here, the sample D was used as an example. All samples were taken from historical objects with the owners’ permission and stored in the laboratory. The sample below comes from Čachtice castle, Slovakia (48.7249653 N, 17.7616083 E). This castle is located about 30 km southwest from the city of Trenčín above the Váh River and was built around the 16th century. It is presumed that the sample is a lime plaster.

#### 3.3.2. Spectroscopy Measurements

The sample was documented using an ordinary mobile phone camera (Figure 16) and a digital DinoLite microscope with magnification 50× (Figure 17) and 200× (Figure 18).

Acquired reflectance spectroscopy data were processed and analyzed using a MATLAB script. Analysis results are mentioned below, and Figure 19 and Figure 20 show spectral graphs of the sample.

Minimum standard deviation: 3.10% (for λ = 1956.520 nm)

Maximum standard deviation: 5.75% (for λ = 1014.341 nm)

Mean standard deviation: 4.00%

#### 3.3.3. Spectroscopy Results

Table 1 shows the representation of the first 10 materials derived from all methods according to their ranking. The average is an arithmetic mean of their determination ranking; standard deviation is also shown in the table to provide a level of confidence. Results of individual analysis can be found at http://hdl.handle.net/10467/94180 (accessed on 14 November 2021).

Spectral curve description.

The curve is flat with reflectance around 60% and small absorption at 1450 nm and more significant one at 1950 nm. The curve is similar to sandstones and lime mortars respectively.

Individual analysis results:SAM—correct detection of lime mortars and sands/sandstones;SID—correct detection of sands/sandstones and dolomite (*CaO*); curve is similar to clay mortar which can show the higher occurrence of *Al*;NNLS—correct detection of lime mortars and sands/sandstones and dolomite (*CaO*); curve is similar to clay mortar which can show the higher occurrence of *Al*;SFF—correct detection of lime mortars and sands/sandstones; curve is similar to clay mortar which can show the higher occurrence of *Al*;BE—correct detection of lime mortars and sands/sandstones;Average—correct detection of lime mortars and sands/sandstones with acceptable standard deviation values.

Conclusion

The sample consists of lime (*CaO*) and quartz sands (*SiO_2_*) with a higher quantity of *Al* and lower of *K*; detection satisfactory.

#### 3.3.4. Scanning Electronic Microscope Findings

Dominant phase:Phase 1—light grey grains—aggregated *SiO_2_* (Al minor occurrence);Phase 2—light grey grains—aggregated *SiO_2_* with *K_2_O* and *Al_2_O_3_* additives; *Na* minor occurrence;Phase 3—white matrix—predominantly *CaO*; *Al*, *Si*, *K* and *Mg* minor occurrence;Phase 4—light grey matrix—predominantly *SiO_2_* with *CaO* and *Al_2_O_3_* additives; *K*, *Fe*, *Na* and *Mg* minor occurrence;

Minor phase:Phase 5—light grey matrix with higher *CaO* percentage and Fe oxides—undistinguishable from phase 4; *Al*, *Si*, *Mg* and *Ti* minor occurrence;Phase 6—dark grey areas—epoxy resin (*C_21_H_25_ClO_5_)*.

Figure 21, Figure 22 and Figure 23 show the sample photos. SEM results (average atomic concentration, average weight concentration and average stoichiometry weight concentration) in the form of graphs are shown in Figure 24, Figure 25 and Figure 26.

#### 3.3.5. Results Comparison

Spectroscopy results indicate that sample is assembled from lime (*CaO*) and sand (*SiO_2_*) with smaller Al and K additive. Scanning electronic microscope results show a high presence of *SiO_2_* and *CaO* with additives *K_2_O* and *Al_2_O_3_*. Elements *Al*, *K* and *Mg* are present in a minority. It can be concluded that both methods provide similar results and thus the detection can be considered as satisfactory.

### 3.4. Plaster Analysis Evaluation Results

Eleven plasters were analyzed using spectroscopy and scanning electron microscope techniques (Section 3.3) in order to find possibilities of reflectance spectroscopy and its ability to determine the composition of plasters and mortars. It was found that reflectance spectroscopy can provide powerful information, but results must be interpreted with care and they are not unequivocal.

Lime mortars’ spectral signature has high vicinity among all four samples and hence the differentiation is lower. Geopolymer is often interchanged with gypsum and vice versa. Since water absorption features were not removed from the spectra, gypsum and geopolymer are often incorrectly mentioned in the material decomposition. The influence of these spectral features is strongly material dependent and provides additional information about materials and samples. The majority of samples consists of sand and lime mixture with different percentage of each. These small differences were not always detected and this is the main limitation of the spectroscopy method. Analysis using continuum removal [38] was also tested but not further used. It is more of a visualization method that highlights extremes and hence can be compared to the histogram stretching [39] method used for remote-sensing data.

Table 2 shows the minimum, maximum and average standard deviation of each sample. The quality of spectroscopy detection depends mainly on the correctness of detected material; standard deviation is a derivative outcome. The average standard deviation lies between 1.80% (Sample A) to 7.00% (Sample E). The high standard deviation of Sample E is caused by the fact that it is assembled by different materials that have variable spectral signatures and are very inhomogeneous. An average standard deviation of all samples was equal to 3.99% and could be found to be sufficient.

### 3.5. Decision Tree

Next to spectroscopy analysis, a decision tree approach was also tested. This method, commonly used for multispectral remote-sensing data [40], chooses previously defined thresholds that should be fulfilled to categorize sample into a specific previously defined class. With the spectroscopy data, the possibilities on threshold definition rise and can become very precise and point at specific attributes of spectral curves.

#### 3.5.1. Class Definition

For this analysis, eight classes were set, namely—Geopolymer, Gypsum, Lime mortars, Limes, Marlstone, Sandstone, Quartz and Unidentified. These classes were chosen based on the CTU Material Spectral Library (Section 3.1) and individual characteristics of their spectral signatures. All 20 materials were divided into these eight classes.

#### 3.5.2. Threshold Settings

To classify each material into the correct class, seven thresholds had to be set. These limits were chosen concerning the nature of each material’s individual spectral signature. Maximum and minimum values in the specific spectral range, as well as the reflectance, were considered. These boundaries were then tuned using the CTU Material Spectral Library to ensure that all material present in the library will be present in the correct class. This was an issue with clay mortar (no. 20), that was integrated into the “Other” class at first, but then it was found that due to the high amount of quartz sand present in the mixture it cannot be spectrally distinguished from the “Quartz” class and, therefore, it was moved and the classes were set as mentioned in Section 3.3.1. If a sample does not fulfil any threshold, it will be included in the “Unidentified“ class.

#### 3.5.3. Data Processing and Results

Data were processed using MATLAB script and the flowchart is shown in Figure 27. Results are expressed in the form of a text file.

When results were compared to outcomes derived from the scanning electron microscope available for every sample it can be concluded that the decision tree provides relevant results. Samples A, C, D and 2 were assigned to the class “Sandstone” since many sand particles are included in the mixture. Samples FA, 1 and 3 were assigned to the class “Lime mortar” which corresponds with scanning electron microscope findings. Sample B was correctly included in the “Gypsum” class and Sample 4 matched the “Lime” class since it is pure CaO on wood. Sample E was classified as “Geopolymer” which is the only incorrect result and was caused by the fact that the sample was very inhomogeneous with visible water absorption spectral bands that are similar to geopolymers.

## 4. Discussion

The goal of this work was to verify the possibilities of non-invasive spectroscopy analysis used for mortar and plaster material determination. As shown by the research, material characteristics can be determined but this technology has its limits. For the spectroscopy analysis it is essential to create convenient spectral library. Based on material samples provided, a CTU Material Spectra Library was created and it is available online upon request from the Laboratory of Photogrammetry, Department of Geomatics, Czech Technical University in Prague, web page (http://lfgm.fsv.cvut.cz/CTU.html, accessed on 14 November 2021). This library includes 20 materials that were chosen to define historical plaster and mortar composition used in the area of Central Europe (the Czech Republic). The spectral library contains information about basic mortar and plaster components like filler (sand, clay) and binder (lime, gypsum, cement) types; however, spectral libraries are geographically limited and only apply to a certain area.

It is also necessary to find a suitable environment for measured sample analysis. Measured unknown sample spectral curves are compared with spectral library data and the result is either in accordance with one of the known material’s spectral curve or more likely a partial correspondence. The spectral curve similarity rate determines the probability of the correct material composition determination. Software for hyperspectral analysis of remotely sensed data provides a convenient environment, and ENVI software was used for this work. The analysis was also performed in other software tools like MATLAB script language, open-source QSdata software, and ENVI software. The biggest benefit of the script language and the opensource software is that it is visible how the variables are computed and parameters can be adjusted. An ENVI software tool has a user-friendly interface and some analytical explanations can be found online, but full control over the processing chain is not enabled. Due to the aforementioned factors, the best computing software cannot be derived and was not the main purpose of this paper.

However, the spectral curve similarity analysis of known materials and unknown samples is not always plausible in itself. It was necessary to find another, but physically completely different, method that would independently verify the results. This can be a detailed chemical analysis or another method based on an electron microscope. In general, these methods are slow and expensive, so that is why there is an effort to determine the composition of plaster and mortar samples relatively cheaply and quickly. In terms of evaluating our approach, we can say that we succeeded in defining the general material composition of the measured samples. The CTU library was tested on 11 plaster and mortar samples. To verify this method, scanning electron microscope results were confirmed to determine the composition of individual materials and samples. It was found that reflectance spectroscopy can be a powerful tool for plaster and mortar material determination when used with care and vigilance. In general, test samples were determined correctly, the majority of samples consists of sand (SiO2) and lime (CaO) mixture with a different percentage of each. These small differences were not always detected and this is the main limitation of this method. However, it is very easy to analyze, for example, samples containing gypsum, where lime was (and still is) not used, as proven by the Iraq mortar sample.

The next goal of the research was to create an automatic means of sample recognition and, therefore a decision tree approach for material composition detection of plaster and mortar samples was also tested (Section 3.5). Materials were separated into eight classes and these thresholds were then applied to plaster and mortar samples. The results show the fine determination of samples into specific classes that correspond to results given by the scanning electron microscope and spectroscopy analysis.

The power of the reflectance spectroscopy method lies in its repeatability that is assured by the calibration procedure. The CTU Material Spectral Library can be used by various researchers worldwide, but has a local validity in general. If a different reflectance spectroscopy device is employed, the user selects the spectral range that corresponds to the operating range of the device used. Its limitation is the spectral range (1000–2500 nm)—and it may be that, due to a small overlap between the spectral range of the CTU Material Spectral Library and the users’ device, there might be a restraint in the application. This can be solved by enlarging the spectral library spectral range with a new device.

In future research, the authors would like to focus on enlarging the material spectral library for a more specific material determination not only of plasters and mortars but also of other historical building materials displaying higher variability.

## 5. Conclusions

Reflectance spectroscopy was used to derive plaster and mortar material composition based on the material spectral library created. It consists of 20 samples and it is accessible via DSpace library (Section 3.1). It is a unique set of material spectral information that is directed at Central Europe and has become the basis for future enlargement. Various analyses were performed and compared.

It was found that the spectroscopy method is suitable for use in the historical heritage documentation field. However, it is not expected to obtain precise results regarding the material composition of man-made mixtures (mortars, plasters) using reflectance spectroscopy. Many mixtures contain similar additives and/or the additives are very alike, and sand is of various quality and composition. European mortars traditionally contain lime. This method is, therefore, very useful for mineral and rock types in geology; in the cultural heritage field it can be used as an assistive or complementary technology for non-invasive material research. It suitably complements the existing and traditional chemical analysis of materials.

The authors believe that the use of non-invasive cultural heritage object documentation will play a key role in the future. When documenting historical objects, owners will prefer non-invasive techniques since they do not damage the object of interest. Nowadays, the methodologies of these techniques should be established in order to be widely used by the public in days to come. In the near future, this work will be extended and the spectral library will be used on other samples.

## Figures and Tables

**Figure 1 materials-14-07030-f001:**
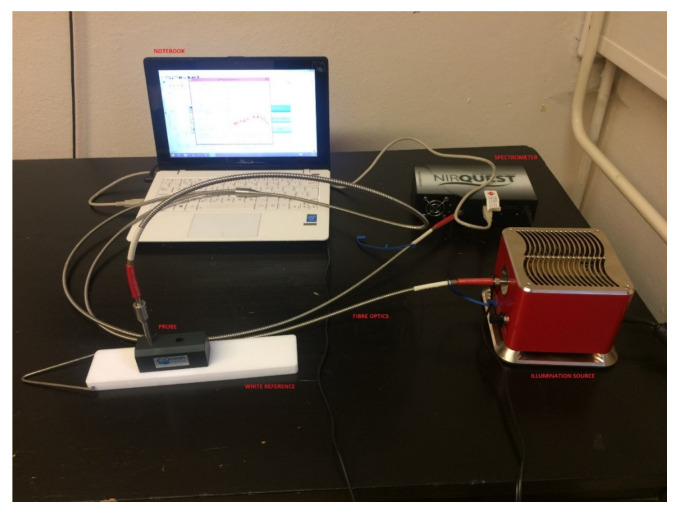
Reflectance spectrometry measuring device NIRQuest512-2.5.

**Figure 2 materials-14-07030-f002:**
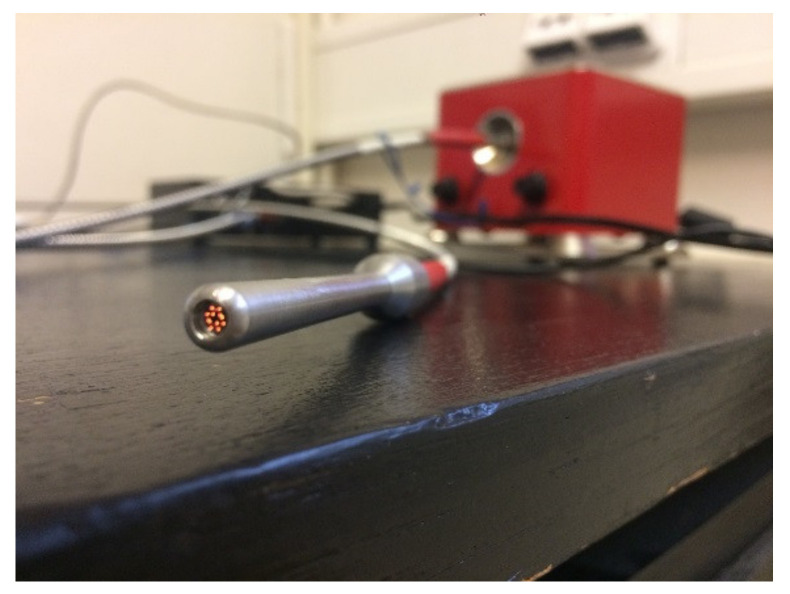
Probe on the end of optical fiber.

**Figure 3 materials-14-07030-f003:**
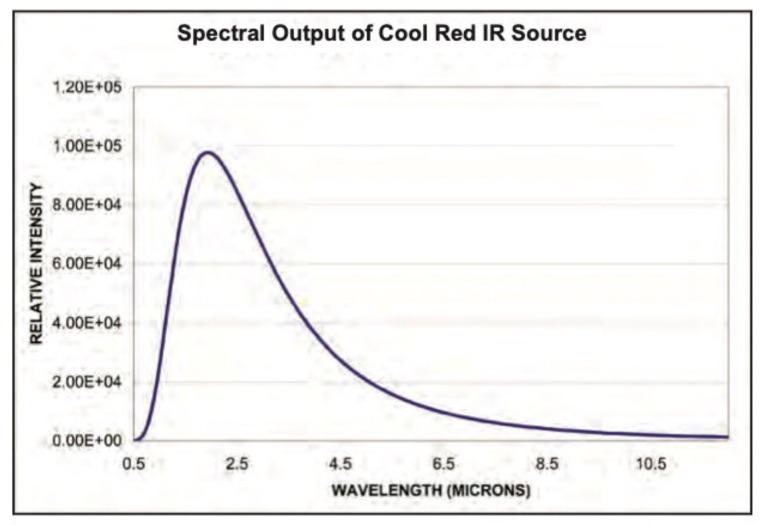
Intensity of Cool Red illumination source and its dependence on wavelength.

**Figure 4 materials-14-07030-f004:**
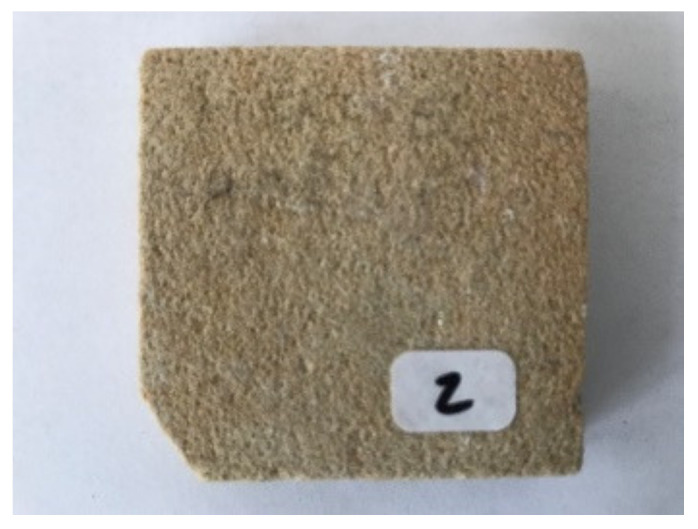
Hořice sandstone sample image.

**Figure 5 materials-14-07030-f005:**
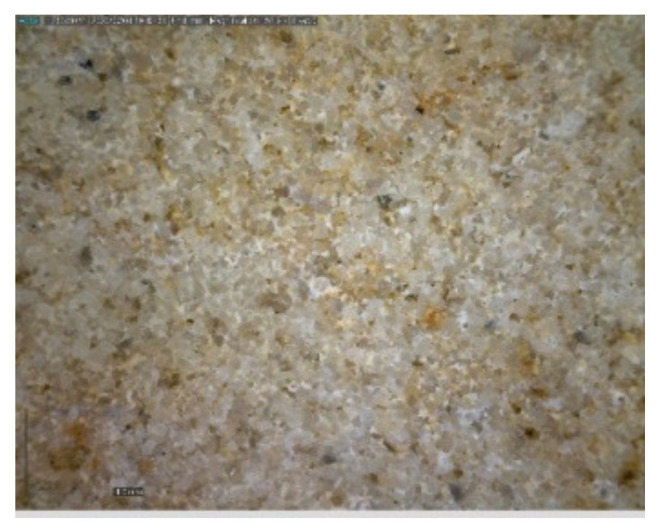
Hořice sandstone DinoLite microscope image, magnification 50×.

**Figure 6 materials-14-07030-f006:**
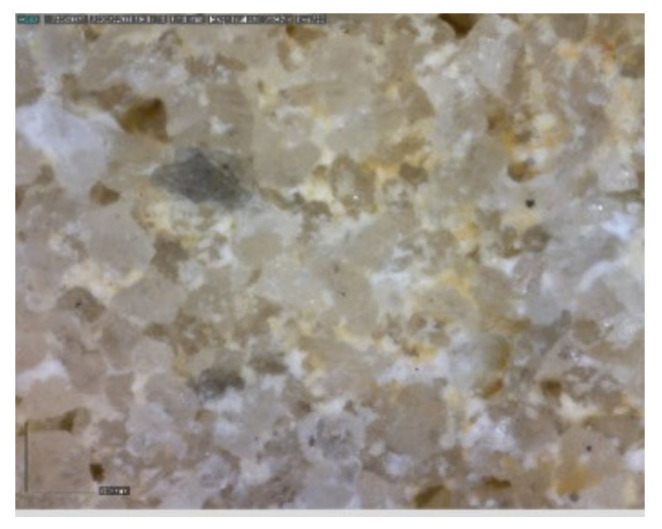
Hořice sandstone DinoLite microscope image, magnification 200×.

**Figure 7 materials-14-07030-f007:**
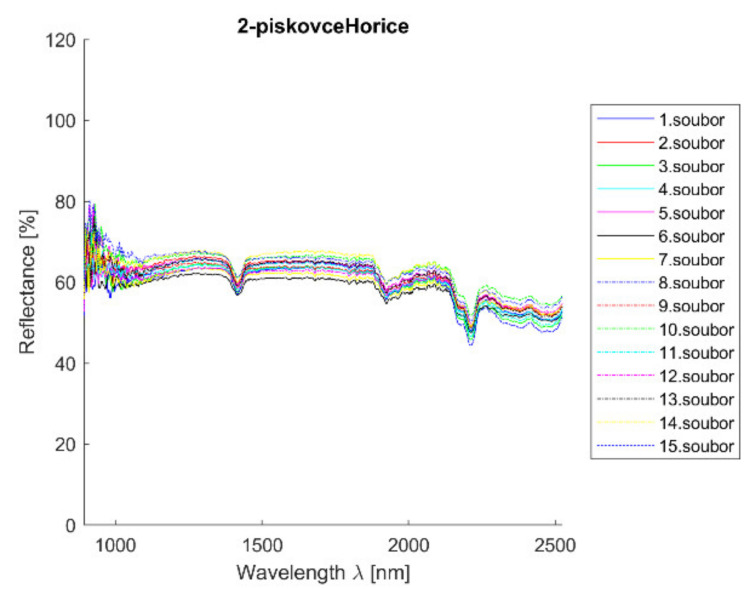
Hořice sandstone—all measurements plot. Measurements were taken uniformly all over the sample to cover all possible sample deviations.

**Figure 8 materials-14-07030-f008:**
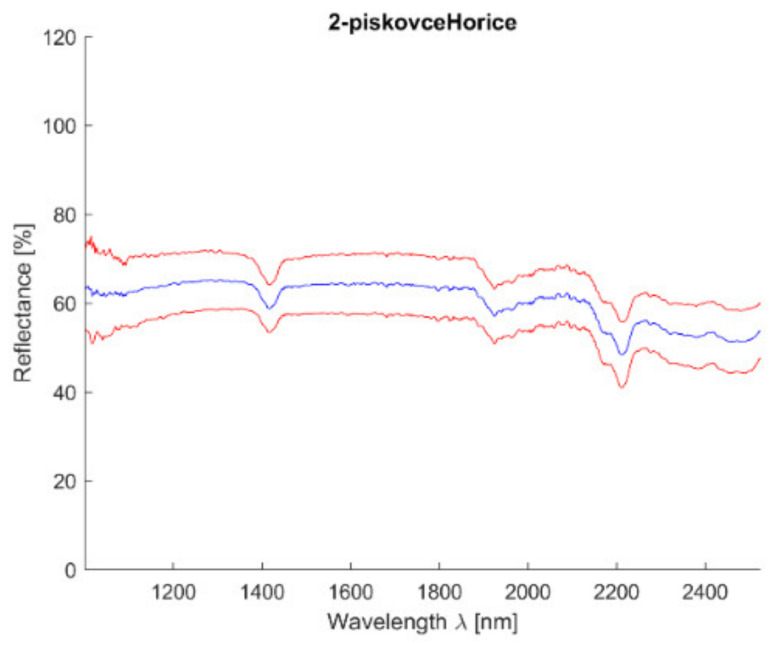
Hořice sandstone—mean value (blue) and 2,5*standard deviation (red).

**Figure 9 materials-14-07030-f009:**
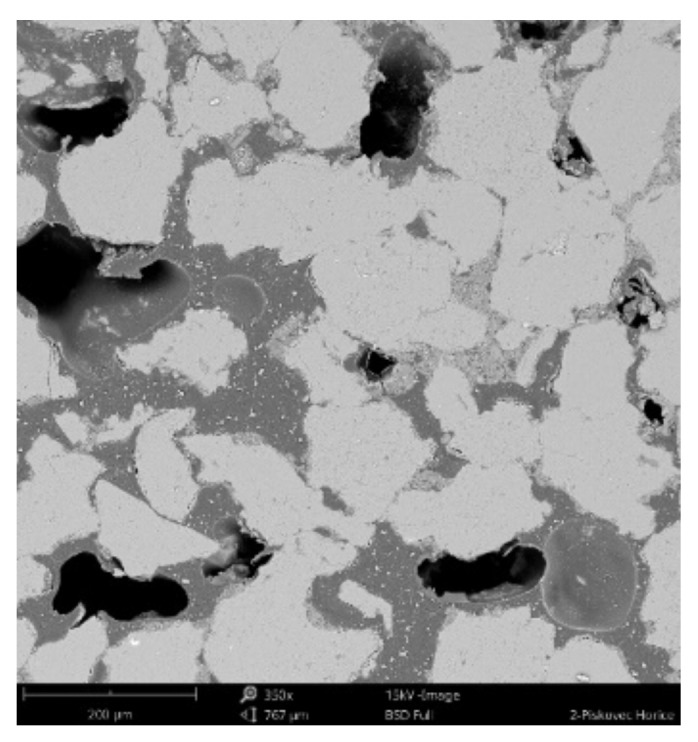
Phenom XL Desktop scanning electron microscope (SEM) photos.

**Figure 10 materials-14-07030-f010:**
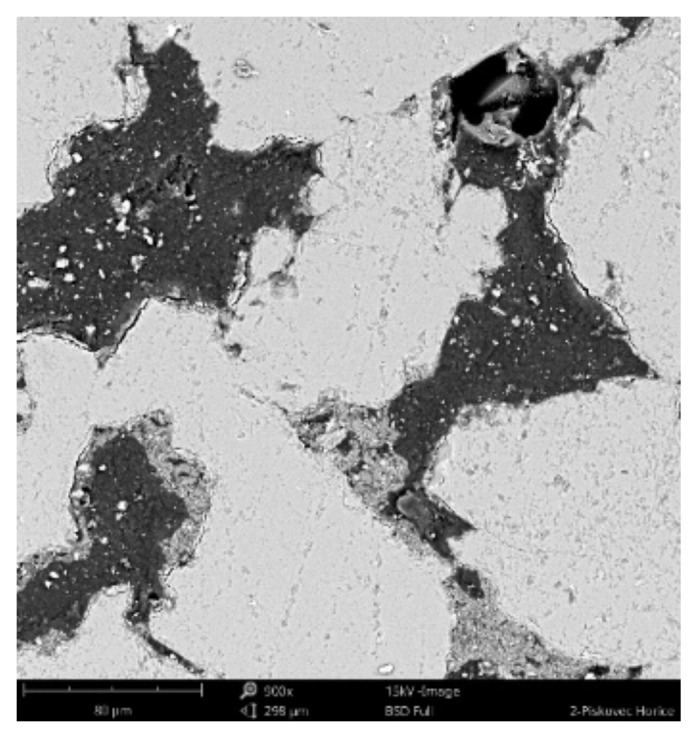
Phenom XL Desktop SEM photos.

**Figure 11 materials-14-07030-f011:**
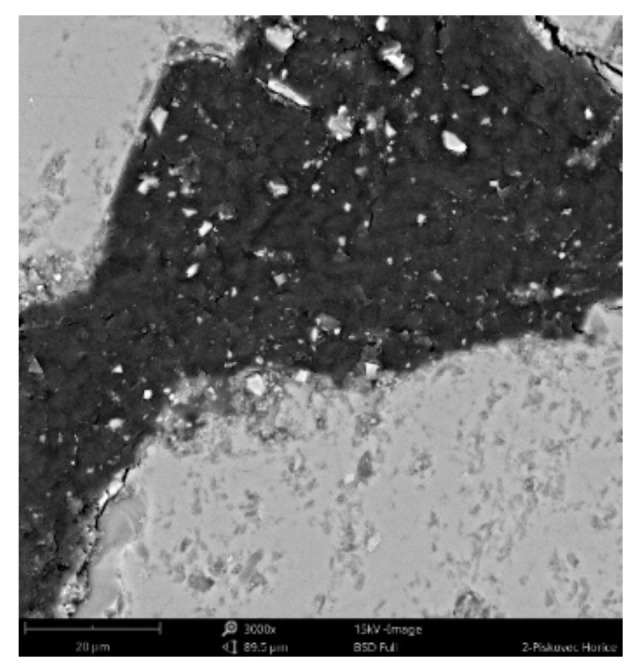
Phenom XL Desktop SEM photos.

**Figure 12 materials-14-07030-f012:**
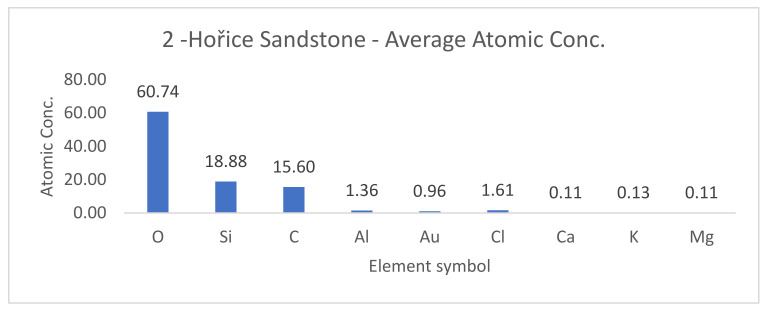
Hořice sandstone average atomic concentration [%].

**Figure 13 materials-14-07030-f013:**
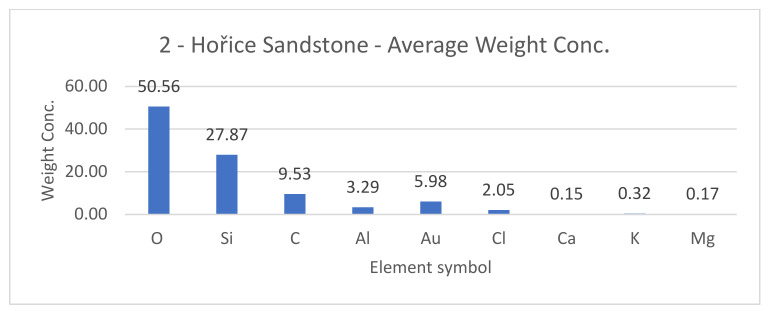
Hořice sandstone average weight concentration [%].

**Figure 14 materials-14-07030-f014:**
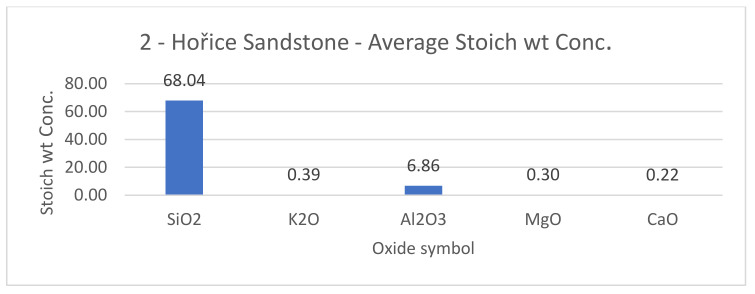
Hořice sandstone average stoichiometric wt concentration.

**Figure 15 materials-14-07030-f015:**
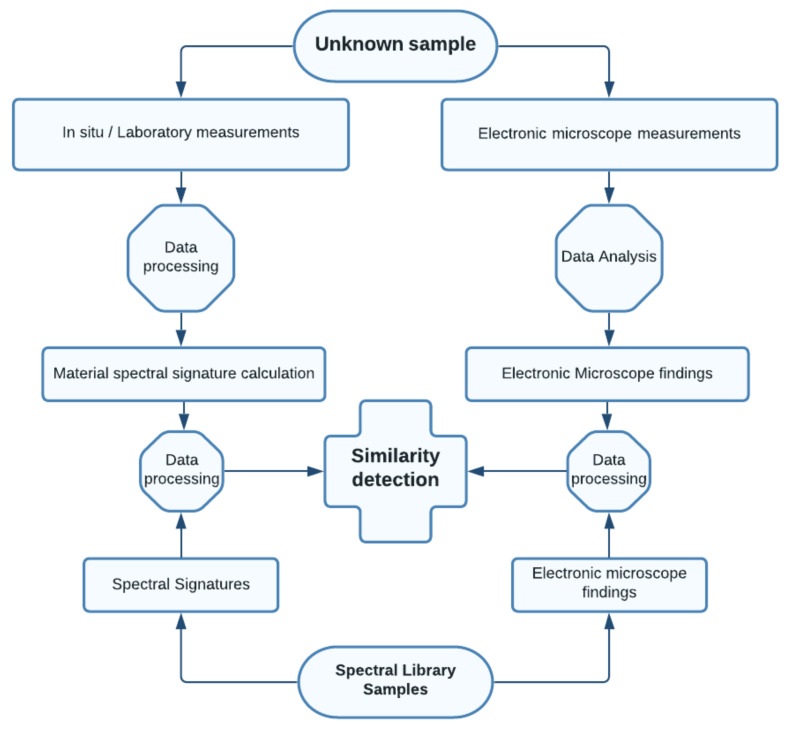
Determination of unknown sample—processing workflow.

**Figure 16 materials-14-07030-f016:**
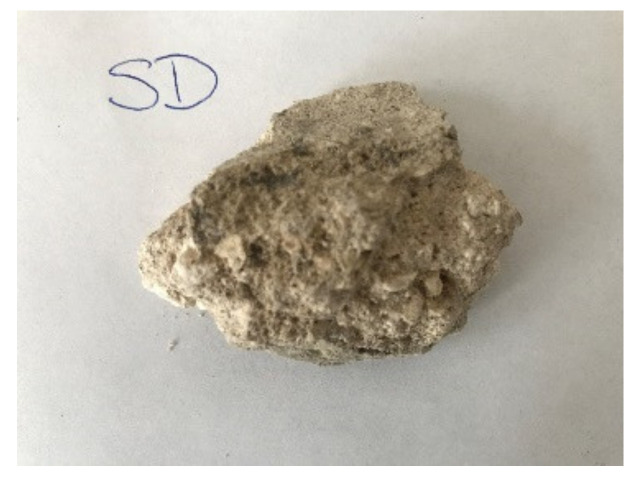
Sample D—sample image.

**Figure 17 materials-14-07030-f017:**
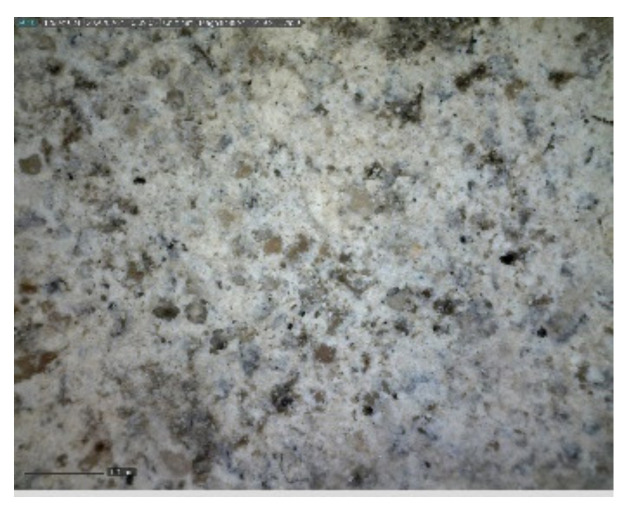
Sample D—DinoLite microscope image, magnification 50×.

**Figure 18 materials-14-07030-f018:**
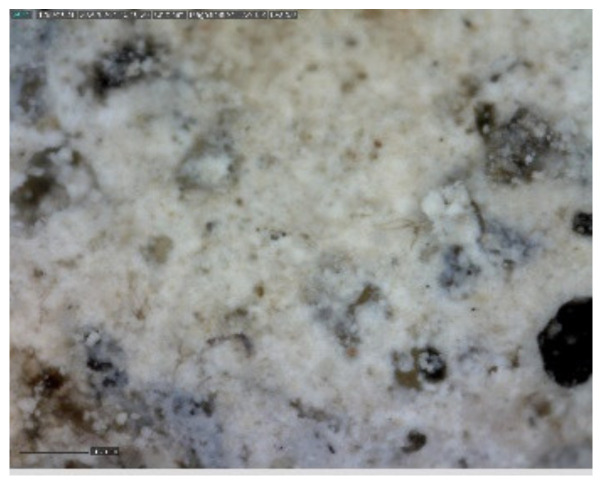
Sample D—DinoLite microscope image, magnification 200×.

**Figure 19 materials-14-07030-f019:**
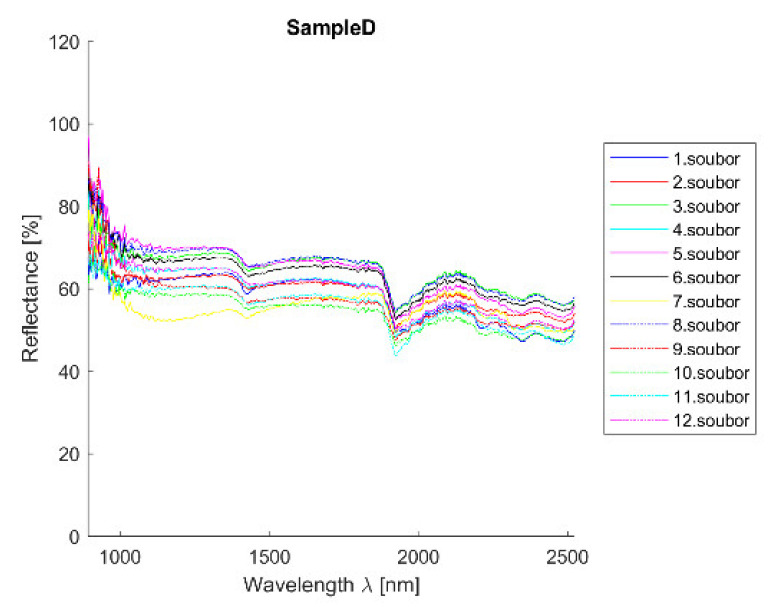
Sample D—all measurements plot. Measurements were taken uniformly all over the sample to cover all possible sample deviations.

**Figure 20 materials-14-07030-f020:**
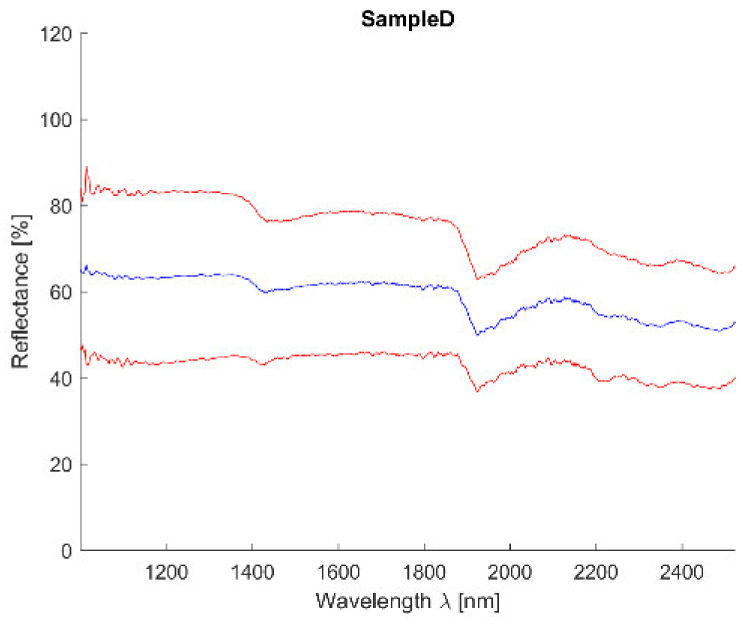
Sample D mean value (blue) value (blue) and 2,5*standard deviation (red).

**Figure 21 materials-14-07030-f021:**
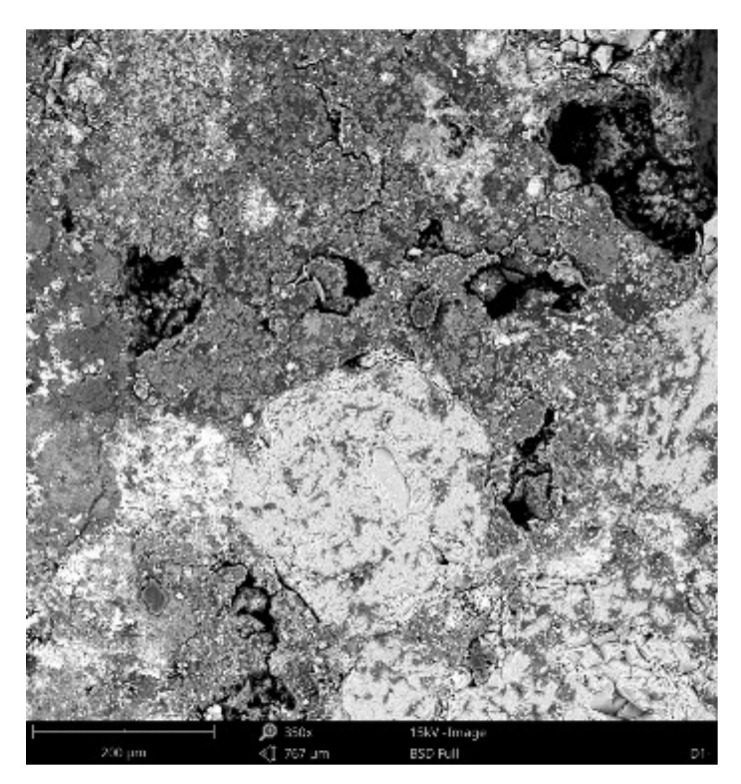
Phenom XL Desktop SEM photos.

**Figure 22 materials-14-07030-f022:**
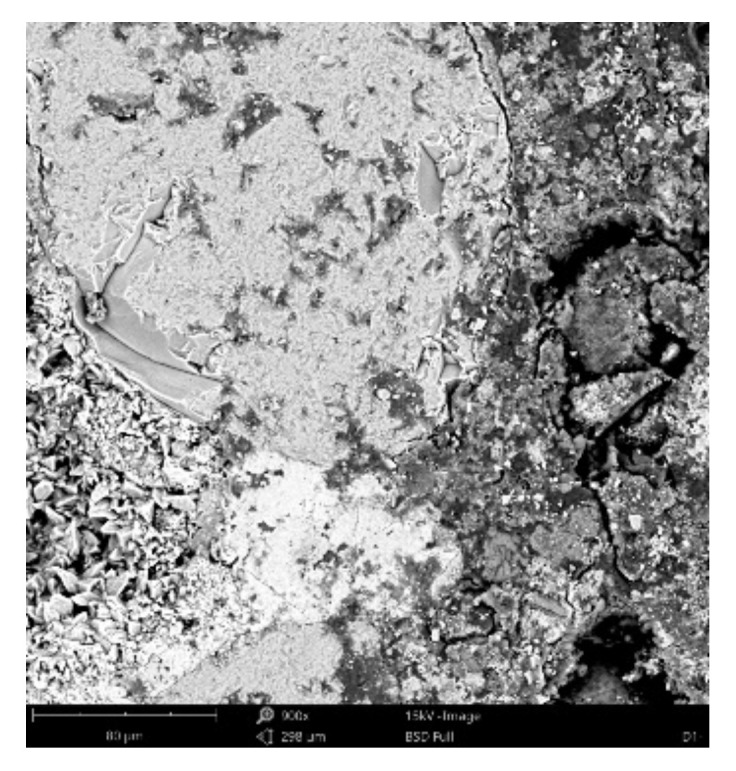
Phenom XL Desktop SEM photos.

**Figure 23 materials-14-07030-f023:**
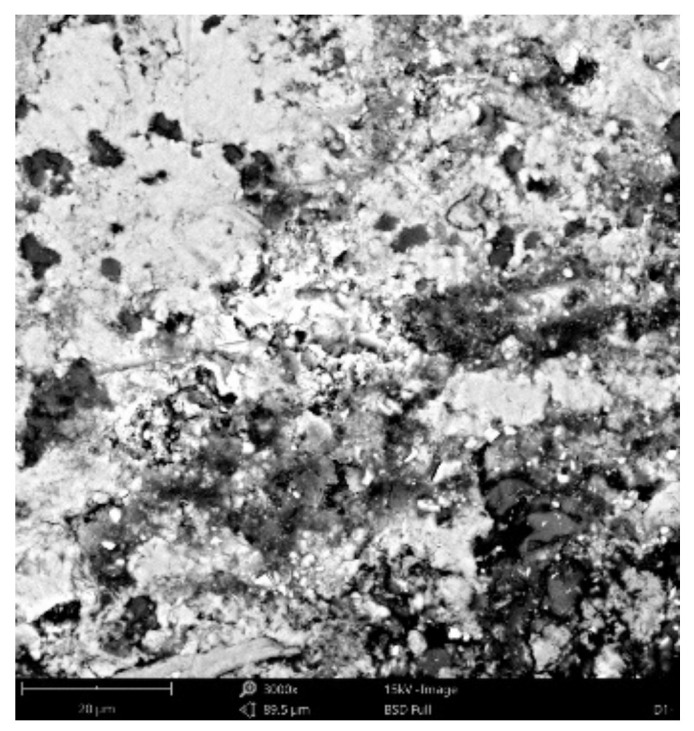
Phenom XL Desktop SEM photos.

**Figure 24 materials-14-07030-f024:**
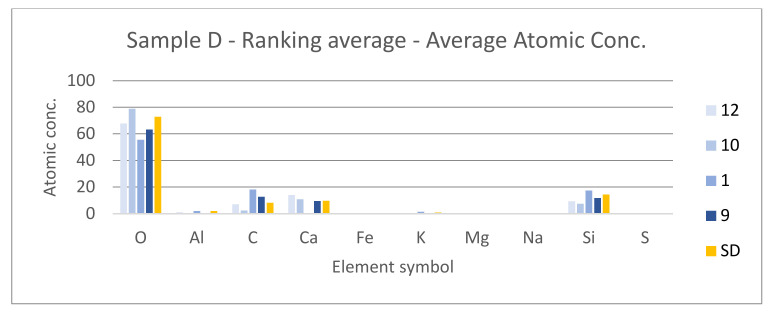
Sample D—ranking average—average atomic concentration.

**Figure 25 materials-14-07030-f025:**
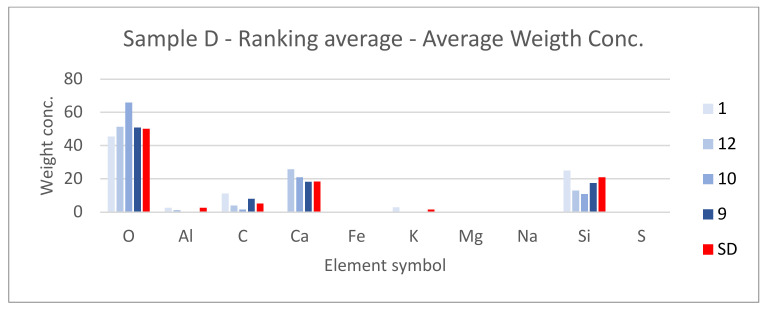
Sample D—ranking average—average weight concentration.

**Figure 26 materials-14-07030-f026:**
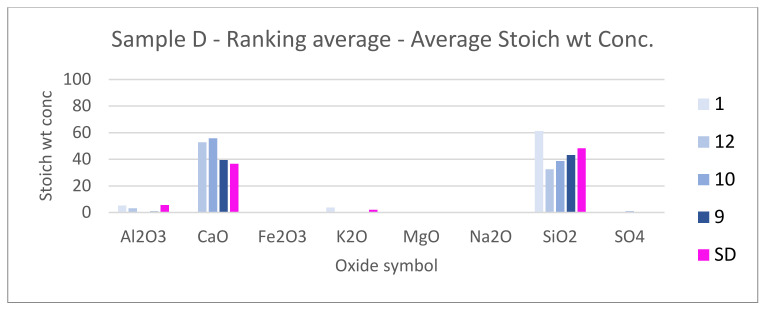
Sample D—ranking average—average stoich wt concentration.

**Figure 27 materials-14-07030-f027:**
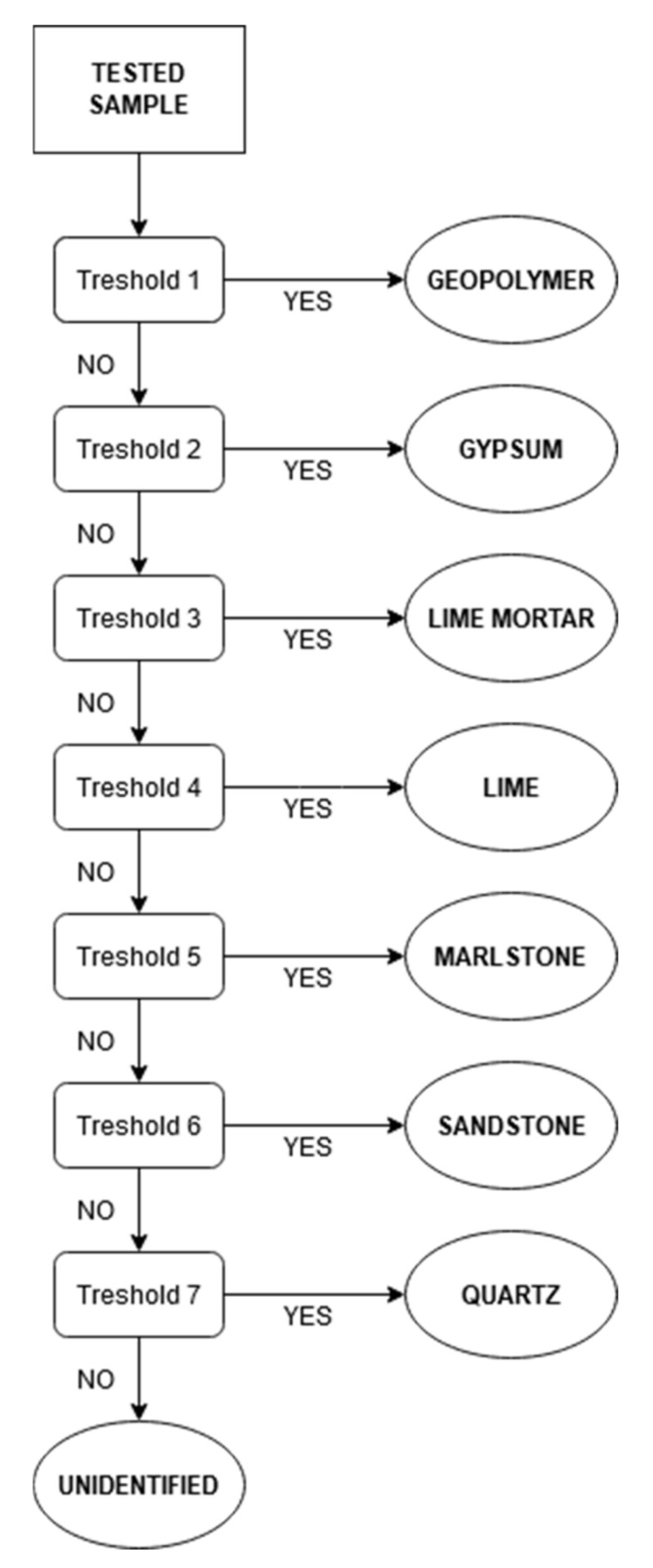
Decision tree flowchart.

**Table 1 materials-14-07030-t001:** Sample D—average material ranking in processed analysis.

Material	Average	Std Dev
12-Lime + Metakaolin Binder Mortar	2.3	1.5
10-Lime + Cement Binder Mortar	4.5	3.4
1-Božanov Sandstone	4.6	2.1
9-Air Lime Mortar	5.0	2.6
11-Hydraulic Lime Mortar (NHL5)	6.5	7.0
2-Hořice Sandstone	7.0	6.1
4-Quartzite	7.7	7.6
15-Borek River Sand	8.3	3.8
6-Přední Kopanina Marlstone	8.6	2.1
5-Maastricht Limestone	9.0	5.2

**Table 2 materials-14-07030-t002:** Sample measurements—data and detection quality.

Sample	Standard Deviation [%]			Spectroscopy Detection Quality
	Minimum	Maximum	Average	
1	1.89	6.42	4.15	Satisfactory
2	3.08	5.05	3.81	Satisfactory
3	2.50	7.69	3.77	Partially satisfactory
4	2.86	5.49	3.80	Satisfactory
A	1.59	3.31	1.80	Nearly satisfactory
B	1.96	5.56	3.13	Nearly satisfactory
C	3.34	9.03	4.47	Nearly satisfactory
D	3.10	5.75	4.00	Satisfactory
E	5.17	8.16	7.00	Partially satisfactory
FA	2.82	4.51	3.86	Satisfactory
FB	3.27	5.65	4.76	Satisfactory
Rock 1	3.06	3.95	3.27	Satisfactory

## Data Availability

The data presented in this study are available on request from the corresponding author.

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
