# Peer review of "Creating a Material Spectral Library for Plaster and Mortar Material Determination"

_materials, 2021, doi:10.3390/ma14227030_

Round 1

Reviewer 1 Report

Information is in the attached file.

Author Response

Dear reviewer,

  please find answers to your comments in attachment. Thank you

Kind Regards,

Eva Matoušková

Reviewer 2 Report

Although the subject could be interesting, the methodology applied is not well described from my point of view. Some data regarding the methodology are lost and some other details must be mentioned.

English must be revised by a native speaker. 

- Lines 42-44: Please, explain better the Raman phenomena

- Lines 47-79: From the best of my knowledge, X-rays usually do not harm the objects of interest

- In the introduction, data about other databases must be mentioned

- Line 83 and Figure 1: Please, mark each of the components in Figure

- Line 94: Figure 2?

- Line 97: Figures 16 and 17 are mentioned before Figures 3-15? 

- Lines 139-141: The information shown in this paper has been published before?

- Line 149: Reference [139]?

- Line 177: Specify the composition of the epoxy resin

- Line 186: SEM-EDX quantification using ZAF? which method was employed?

- Lines 246-255: What are the meaning of SAM, SID, NNLS, SFF and BE? Some data about the methodology of the comparison between unknown samples and databases must be more profusely and clearly explained. 

- Section 3.3 Sample D is similar to samples 1, 9, 10 and 12?

- Line 323: Section 3.5.?

- Line 395: Thesis or paper? The paper seems to be a transcription of a chapter of a thesis dissertation

- Discussion must be notably improved

-  

Author Response

Dear reviewer,

please find responses to your comments in attachment.

Kind regards,

Eva Matoušková

Reviewer 3 Report

The work titled "Creating a Material Spectral Library for Plaster and Mortar Material Determination" needs major modification. Some comments and recommendations are provided below:

1 - Authors could put '"SEM" for Electron Microscope and EDS for energy-dispersive spectrometer.

2 - the word façade is not in the english language.

3 - The sentence say: " These methods are very complex and due to the nature of these methods sample heating can be a big issue". Sample heating does not happen in low power raman spectroscopy.

4 - Improve the sentece "These patterns of reflectance and absorption across wavelengths " . Change term across, could switch to depending on the wavelength.

5 - line 67 "but without specialization" explain?

6 -  What is the power of the lamp used. describe more details about the 400nm to 1000nm region (optical power as well). It has few details throughout the text.

7 - line 97 - Figure 17 will be used as a white reference is correct?

8 -  put trademark for ocean optics

9 -  could better write the methodology highlighting the background of the measurements and identification of the samples.

10 - could change magnification 50 for magnification 50x

11 - the description in figure 6 is very poor. could improve.

12 - details of the identification of the microscopy (figure 7,8,9) should be removed and only the scale placed

13 - geographic position of the samples by latitude. Improve subtitles. The identification and analysis of the samples in section 3.3 must be redone.

Author Response

(The authors gave the same response as above.)

Round 2

Reviewer 2 Report

-

Author Response

Dear reviewer,

english language has been reviewed and updated, thank you for your comments.

Kind regards,

Eva Matoušková

Reviewer 3 Report

The authors have addressed all the questions, but I suggest it publicaiton after one minor revision.

reflectance is capitalized in some cases but not in others

line 47 shoud chance historic to historical.

fibre in figure 2 , what meaning?

line 22 put "scanning" electronic microscope (SEM)

please define SEM only one once.

in line 47 you should change the word "surface" to "material" or "sample . Because in  raman spectroscopy its possible 
measure with deep profile 

in line 102 I did not understand. improve this expression :"the material is Silicon Nitride (Si3N4) with temperature 1500°K."

Are some figures of the methodology the same as in article HISTORICAL PLASTER COMPOSITION DETECTION USING REFLECTANCE SPECTROSCOPY DOI: 10.14311/CEJ.2016.04.0023? this could give problems related to plagiarism action

in line 143" for more information see Chapter 2.1.." chapter 2.1????

on line 225 it has two commas

Author Response

Dear reviewer,

  thank you for your comments, the manuscript has been updated accordingly. Regarding copyright of used figures there will be no issues since they were created by myself (the corresponding author). 

Kind Regards,

Eva Matoušková
